# *MMP2* Polymorphism Affects Plasma Matrix Metalloproteinase (MMP)-2 Levels, and Correlates with the Decline in Lung Function in Hypersensitivity Pneumonitis Positive to Autoantibodies Patients

**DOI:** 10.3390/biom9100574

**Published:** 2019-10-05

**Authors:** Luis Santiago-Ruiz, Ivette Buendía-Roldán, Gloria Pérez-Rubio, Enrique Ambrocio-Ortiz, Mayra Mejía, Martha Montaño, Ramcés Falfán-Valencia

**Affiliations:** 1Interstitial Lung Disease and Rheumatology Unit, Instituto Nacional de Enfermedades Respiratorias Ismael Cosío Villegas, Mexico City 14080, Mexico; luissantiago091@gmail.com (L.S.-R.);; 2Translational Research Laboratory on Aging and Pulmonary Fibrosis, Instituto Nacional de Enfermedades Respiratorias Ismael Cosío Villegas, Mexico City 14080, Mexico; ivettebu@yahoo.com.mx; 3HLA Laboratory, Instituto Nacional de Enfermedades Respiratorias Ismael Cosío Villegas, Mexico City 14080, Mexico; glofos@yahoo.com.mx (G.P.-R.); e_ambrocio@iner.gob.mx (E.A.-O.); 4Department of Research in Pulmonary Fibrosis, Instituto Nacional de Enfermedades Respiratorias Ismael Cosío Villegas, Mexico City 14080, Mexico; mamora572002@yahoo.com.mx

**Keywords:** hypersensitivity pneumonitis, metalloproteinases, genetic association, autoantibodies, MMP1, MMP2, SNPs

## Abstract

Among hypersensitivity pneumonitis (HP) patients have been identified who develop autoantibodies with and without clinical manifestations of autoimmune disease. Genetic factors involved in this process and the effect of these autoantibodies on the clinical phenotype are unknown. Matrix metalloproteinases (MMPs) have an important role in architecture and pulmonary remodeling. The aim of our study was to identify polymorphisms in the *MMP1*, *MMP2*, *MMP9* and *MMP12* genes associated with susceptibility to HP with the presence of autoantibodies (HPAbs+). Using the dominant model of genetic association, comparisons were made between three groups. For rs7125062 in *MMP1* (CC vs. CT+TT), we found an association when comparing groups of patients with healthy controls: HPAbs+ vs. HC (*p* < 0.001, OR = 10.62, CI 95% = 4.34–25.96); HP vs. HC (*p* < 0.001, OR = 7.85, 95% CI 95% = 4.54–13.57). This rs11646643 in *MMP2* shows a difference in the HPAbs+ group by the dominant genetic model GG vs. GA+AA, (*p* = 0.001, OR = 8.11, CI 95% = 1.83–35.84). In the linear regression analysis, rs11646643 was associated with a difference in basal forced vital capacity (FVC)/12 months (*p* = 0.013, β = 0.228, 95% CI95% = 1.97–16.72). We identified single-nucleotide polymorphisms (SNPs) associated with the risk of developing HP, and with the evolution towards the phenotype with the presence of autoantibodies. Also, to the decrease in plasma MMP-2 levels.

## 1. Introduction

Hypersensitivity pneumonitis (HP) is a complex disease caused by an exaggerated immune response to the inhalation of a wide variety of organic particles [1]. Although it has been established that the development of the disease depends on the time of exposure and the antigenic load, only a small proportion of individuals exposed to antigens associated with HP develop the disease, suggesting additional host and environmental factors may play a role in the pathogenesis [2].

In chronic stages, HP is characterized by progressive lung remodeling, which is associated with the loss of functional architecture and an excessive extracellular matrix (ECM) deposition [3]. Therefore, ECM proteins have been previously explored as indicators of disease activity, and as potential biomarkers of diagnosis and prognosis in patients with chronic obstructive pulmonary disease (COPD) and idiopathic pulmonary fibrosis (IPF) [4].

Recently, the existence of hypersensitivity pneumonitis with autoimmune characteristics (HPAF) in a US cohort has been described, determining the prevalence of 15% in the HP patients; in addition, the autoimmunity profile was recognized as an independent predictor of mortality [5]. It is not clear if HPAF patients have different genetic susceptibility characteristics to the non-autoimmunity HP patients. So far there are no genetic studies on the transforming phenotype of HP patients positive to autoantibodies.

Our aim was to identify single nucleotide polymorphisms (SNPs) associated with genetic susceptibility in metalloproteinases genes (*MMP1, MMP2, MMP9,* and *MMP12*) in HP patients with and without serum autoantibodies, as well as to the progression or development of the disease.

## 2. Materials and Methods

### 2.1. Study Population

An analytical, cross-sectional study was conducted. One hundred and thirty-eight patients with Hypersensitivity Pneumonitis (HP) from the Instituto Nacional de Enfermedades Respiratorias Ismael Cosío Villegas (INER), at Mexico City, Mexico were included.

The diagnosis was established according to criteria based upon the presence of HP, either by high-resolution chest tomography, bronchioloalveolar lavage (BAL) with lymphocytosis ≥ 40% and/or positive avian antigen. Additionally, in cases without a definitive clinical diagnosis, a lung biopsy was performed to confirm the diagnosis. Subjects with positive serology were included for at least one of the following antibodies: Antinuclear antibodies (ANAs) with a specific pattern of connective tissue disease of any kind (cytoplasmic, nucleolar, centromeric); ANA with pattern homogeneous, fine or coarse mottle, with titles ≥ 1:320; at least one antibody in the autoimmunity profiles for myositis or systemic sclerosis; rheumatoid factor ≥ 3 times the lower normal limit; Anti-cyclic citrullinated peptide (anti-CCP) ≥ 20. Salivary gland biopsy (grade 3–4). All of them without classification criteria according to the American Society of Rheumatology (ACR/EULAR) for connective tissue disease (CTD). A healthy subjects reference group with one hundred and eighty-four controls (HC) were included.

This study was approved by the Institutional Committee for Science and Ethics of the Instituto Nacional de Enfermedades Respiratorias Ismael Cosío Villegas (INER) (approbation codes: B20-15 and C60-17).

### 2.2. DNA Extraction

We obtained an 8-mL peripheral blood sample from each participant through venipuncture. Blood was collected in tubes with EDTA as an anticoagulant. The DNA extraction was performed using a BDtract DNA isolation kit (Maxim Biotech, Inc. San Francisco, California, USA) and later was quantified with a NanoDrop 2000 (Thermo Scientific, DE, USA). The contamination with organic compounds and proteins was determined by establishing the ratio of 260/240 and 260/280 readings, respectively. The samples were considered free of contaminants in both cases when the ratio was between 1.7 and 2.0.

### 2.3. Selection of Single Nucleotide Variation

The selection process of the evaluated single-nucleotide polymorphisms (SNPs) included a literature review of previous reports of the genetic association of SNPs in matrix metalloproteinase (MMP) genes associated to respiratory diseases in Caucasian and Asian populations, using the NCBI (National Center for Biotechnology Information) database, and including scientific articles published between 2007 and 2016. 

For the four included genes, tag SNP selection was performed with HaploView version 4.2, using the minor allele frequency (MAF) > 10% and r^2^ ≥ 0.80 in the Caucasian population as a reference. A total of 12 SNPs were selected, and data including chromosome location and polymorphism base change are shown in Table 1.

### 2.4. Genotyping

Alleles and genotypes were determined by real-time PCR, 3 μL of DNA were obtained at a concentration of 15 ng/μL. Under the following conditions: 50 °C for 2 min, 95 °C for 10 min, followed by 40 cycles of 95 °C for 15 seconds, and a final cycle of 60 °C for 1 min. The alleles and genotypes of the SNPs were determined by real-time PCR (Real-time PCR System 7300, Applied Biosystems, CA, USA) by allelic discrimination using TaqMan Probes at a concentration of 20× (Applied Biosystems. Foster City CA. USA). In addition, three controls without template (contamination controls) were included for each plate.

### 2.5. Obtaining Plasma Levels of MMP-2 with ELISA

Based upon the results of the genetic association analysis, plasma MMP-2 protein levels were measured using commercial kits (Elabscience Biotechnology Inc. Houston, TX. USA). Readings were obtained using the iMark™ Microplate Absorbance Reader (Bio-Rad, CA, USA).

### 2.6. Statistical Analysis

The statistics program SPSS v.21 (SPSS Inc., Chicago, IL, USA) was used to describe the study population and determine the median, minimum and maximum values for each variable, and compared using U de Mann-Whitney. Continuous variables were reported as means with standard deviation (SD) and compared using a Student’s *t*-test. Categorical variables were reported as counts and percentages, and compared using Fisher’s exact test.

To determine the SNPs associated with the disease’s risk, the frequencies of the alleles and genotypes of the study groups were compared, and the odds ratio (OR) was calculated with a 95% confidence interval (CI), using Epi Info version 7.1.5.2 (CDC, GA, USA).

Statistical significance was considered if the *p*-value < 0.05. The ancestral allele was used as the reference for each of the polymorphisms, and was included population data for the frequencies of the SNPs studied in the HapMap-MEX (Mexican population residing in Los Angeles, California, USA), from the HapMap project (International HapMap Project).

In addition, a linear regression analysis was conducted to investigate the independent effect of the associated genotypes on lung function: Diffusing lung capacity factor for carbon monoxide (DLCO), basal forced vital capacity (FVC) and a difference in basal FVC/12 months.

MMP-2 levels were compared by median values and interquartile ranges for three comparisons: (1) Genotype, (2) the phenotype with positive and (3) negative to autoantibodies phenotype.

## 3. Results

We included 138 patients with HP for the present study. Thirty-four of these patients had autoantibodies present in serum without classification criteria for CTD. Additionally, a reference group with 184 healthy controls was included. They were paired in gender and age with our corresponding study group. (157 women and 27 men, 54.4 ± 12.78 years of age).

The baseline demographic data and the clinical characteristics of the entire cohort showed that the mean age at the time of HP diagnosis was 51 years (± 11 years); with a BMI of 27 (± 5). Comorbidities included diabetes mellitus (12%) and systemic arterial hypertension (22%). 25% of the patients are former smokers. In the physical examination, the most frequent clinical signs were fever (39%) and digital clubbing (31%). Exposure to the avian antigen (88%) was the most common environmental agent identified.

Comparing the demographic characteristics between HPAbs+ and HP without the presence of autoantibodies (Table 2) both groups were found paired in age and gender. In the HP group, a higher proportion of former smokers was observed (29% vs. 14%, *p* = 0.01), also demonstrating more subjects with systemic hypertension (25.9% vs. 8.8%, *p* = 0.002). There were no differences between the groups with respect to other demographic characteristics and antigen exposure. During the study period, 19% of the entire cohort died. There were no statistically significant differences between the groups with respect to the number of deaths (20% vs. 18%, *p* = 0.9). The number of patients with a decrease of ≥ 10% in the predicted FVC differed between the groups (26.4% in HPAbs+ vs. 5.7% in HP, *p* = 0.0001).

When comparing the respiratory function tests and the clinical laboratory characteristics between both groups (Table 3)**,** those patients with the presence of antibodies (HPAbs+) showed a better FVC predicted and DLCO without being statistically significant.

However, those patients with HP showed differences in relation to PSAP, with a higher median compared to the HPAbs+ group (32 mm Hg vs. 40 mmHg *p* = 0.002) required a greater number of patients with supplemental oxygen (26% vs. 35%; *p* = 0.09).

In the laboratory studies, the greater optical density of avian antigen was identified in the HPAbs+ group (1.45 DO vs. 0.89 DO, *p* = 0.03). C reactive protein was also compared as an acute phase reactant in both groups, showing higher levels in patients with HPAbs+ (1.023 mg/dl vs. 0.541 mg/dl, *p* = 0.006). Regarding bronchoalveolar lavage (BAL), the percentage of lymphocytosis was higher in the HPAbs+ group (54.5% vs. 46.2%, *p* = 0.03).

The most expressed antibodies in our study group (Table 4) were the ANA type (50%) showing a higher frequency in their expression with a homogeneous pattern (20%) ≥ 1:320. Followed by others, such as the rheumatoid factor (14.9%) among the most frequently observed.

### 3.1. Analysis of Association by Alleles and Genotypes

Alleles associated with risk were identified in the *MMP1* and *MMP2* genes. The T allele for rs7125062 was found associated when comparing groups of patients with healthy subjects: HPAbs+ vs. HC (*p* < 0.001, OR = 3.69, CI 95% = 2.16–6.29); HP vs. HC (*p* < 0.001, OR = 2.97, CI 95% = 1.99–4.09). (Appendix A). Regarding rs11646643 in *MMP2*, the allele A in frequency was statistically significant when comparing both groups of patients: HPAbs+ vs. HP (*p* = 0.03, OR = 1.88, CI 95% 1.06–3.33). When compared with healthy subjects, only the HPAbs+ group obtained a statistically significant difference (*p* = 0.01, OR = 2.35, CI = 95% 1.36–4.05). The group of HP patients showed no difference with the reference group of healthy subjects. (Appendix A) Allele and genotype frequencies for the three study groups and HapMap-MEX population are included in the Appendix A.

Using the dominant model of genetic association, comparisons were made between the three groups. For rs7125062 (CC vs. CT + TT) in the *MMP1* gene, an association was found when comparing both groups of patients with healthy subjects, respectively: HPAbs+ vs. HC (*p* < 0.001, OR = 10.62, CI 95% = 4.34–25.96) and HP vs. HC (*p* < 0.001, OR = 7.85, CI 95% = 4.54–13.57). (Table 5).

In addition, grouping all HP patients (HPAbs+ and HP) were compared against HC group, observing an association, (*p* < 0.001, OR = 8.42, CI 95% = 5.07–13.98). (Appendix A).

On the other hand, for rs11646643 in *MMP2*, HPAbs+ patients presented a lower frequency in the homozygous GG genotype compared to the HP group (GF = 5.88% vs. 33.65%). The difference was statistically significant when comparing them based on the dominant genetic model GG vs. GA + AA, (*p* = 0.001, OR = 8.11, CI 95% = 1.83–35.84). A similar effect occurs when comparing the group HPAbs+ vs. HC (*p* < 0.001, OR = 11.51 CI 95% = 2.67–49.49). (Table 5) Interestingly, the association was significative comparing all HP patients (independently of the serological phenotype) against healthy subjects, respectively: HP (all) vs. HC (*p* = 0.006, OR = 1.96, CI 95% = 1.21–3.16). (Appendix A).

For the rest of the SNPs of *MMP1* and *MMP2,* no significant associations were found. There was no association for the *MMP9* and *MMP12* genes. Data for genetic association models for SNPs and genotypes in *MMP9* and *MMP12* genes are shown in Appendix A.

In the linear regression analysis, only rs11646643 (GA+AA) was associated to a difference in basal FVC/12months (*p* = 0.013, *β* = 0.228, 95% CI, 95% = 1.97–16.72). No other SNPs or variables were associated.

The genotype and allele frequencies of the 12 evaluated SNPs in Mexicans residing in Los Angeles (HapMap-Mex) are shown in the Appendix A.

Genotype frequencies of the two associated SNPs in the HC group were compared to those in the HapMap-Mex. The frequency of the rs7125062 CT genotype in the *MMP1* gene in the HC group was reduced when compared with the population data from the HapMap-Mex (2.17% vs. 58.0%), contrary to observed with the frequency of the CC genotype, which was elevated in the HC group. (Appendix A).

For the associated SNP in the *MMP2* gene, (rs11646643) the AA genotype frequency in the HC group was similar from the reported in the HapMap-Mex (32.61% vs. 30.0%). Interestingly, the GG and GA genotype frequencies have important differences in their distribution. (Appendix A).

### 3.2. MMP-2 Plasma Levels

Plasma levels of the MMP-2 protein of the HPAbs+ and HP groups were measured by ELISA using commercial kits. There were no significant differences when comparing the levels between patients’ groups (132.98 vs. 130.06 ng/ml, *p* = 0.8). However, when these groups were merged and grouped by the rs11646643 genotype (GG vs. GA+AA), there was a statistically significant difference (188.84 vs. 123.39 ng/ml, *p* = 0.009), maintaining this tendency in the same comparisons, but this time intragroup. (Figure 1)

## 4. Discussion

In this study we demonstrated that some patients with HP express autoantibodies during clinic evolution without classification criteria for connective tissue disease.

We observed that the presence of autoantibodies establishes an inflammatory phenotype with an elevation of acute phase reactants compared to seronegative HP. However, no differences were observed in respiratory function tests.

There is little knowledge regarding this transforming phenotype with the presence of antibodies present in the serum, as well as the identification of clinical factors associated with poorer results.

In the cohort of patients with HP, most of the subjects were in a chronic phase. The group of subjects with HP (without the presence of antibodies) showed a higher degree of pulmonary hypertension with a decrease in DLCO compared to the seropositive subjects, while in the group with HPAbs+, the levels of avian antigen and C-reactive protein were higher, remarking inflammation and lymphocytosis in the BAL.

We can propose from these differences that the HPAbs+ group develops an inflammatory phenotype, and the presence of autoantibodies could have a detonating role. As in IPF, acute exacerbations can occur in chronic HP conditions, and exposures to inhaled antigens can trigger these exacerbations.

However, despite the avoidance of antigens, progressive pulmonary fibrosis and death may occur, suggesting that additional factors may contribute to the activity of the disease.

In the chronic stages, HP is characterized by a progressive pulmonary remodeling, which is associated with the loss of functional architecture and the excessive deposition of ECM.

Among the metalloproteinases, it has been shown that MMP-1 and MMP-2 play an important role in the remodeling of the respiratory tract, and also are implicated in the degradation of fibrillar collagen, gelatin and collagen type IV, that are the most abundant compounds of the extracellular matrix and the basal membranes [6,7]. Besides, it is known that MMP-1 increases the proliferative and migratory capacities of the epithelial alveolar cells and protects the cells of death [8]. 

Recently some studies prove that MMP-1 and MMP-2 mediate the suppression induced by interleukin-13 of the expression of the mRNA of elastin in the fibroblasts of the respiratory tract. MMP-2 also induces the production of collagen, activating the latent transforming growth factor beta-1, an important factor that helps fibrogenesis [9].

In this study of genetic association, we show that two SNPs (of the twelve evaluated in the *MMP1*, *MMP2*, *MMP9* and *MMP12* genes) are associated with genetic susceptibility for HP in Mexican Mestizo population. In the *MMP1* gene, the rs7125062 was associated in the comparations HPAbs+ and HP vs. HC group; while rs11646643 in the *MMP2* gene is associated with the risk of disease and the decline of the lung function in HP; i.e., we can able to identify genetic differences among HPAbs+ and HP patients without autoantibodies.

The rs7125062 was associated in comparisons between groups by serological phenotypes against healthy subjects (HC); In addition, this association was maintained comparing all patients regardless of the phenotype (positive or negative to autoantibodies) against healthy subjects. Observing a tendency of association in relation to the respiratory disease, and not necessarily to the serological phenotype. Regarding rs11646643 of the *MMP2* gene, it is possibly associated with the serological phenotype of the disease. It is possible to perceive due to the increased risk in the respective comparisons: HPAbs+ vs. HP; HPAbs+ vs. HC, (OR = 8.11, OR = 11.51). However, this does not happen to compare the HP vs. HC group.

The modification of the effect on the risk is observed when comparing all patients with HP (regardless of the serological phenotype) against healthy subjects (*p* = 0.006, OR = 1.96), observing the influence of the seropositive phenotype on this association and therefore the role of this SNP.

Just a little proportion of the exposed individuals to an antigen associated with HP will develop the disease, suggesting the existence of another genetic factor associated with the risk. Nowadays there are no studies of the genetic association of rs7125062 and rs11646643 with HP. Therefore, this is the first report of genetic association that involves polymorphism of a single nucleotide with the risk of developing and decline of lung function in the Mexican mestizo population.

In the last SNP, it is located at the GATA-1 (CTATCT) site of the promoter region of the *MMP2* gene [10], that has two sites of the union for the transcription factors AP-2, p53, Sp1 y Sp3 [11]. These transcription factors regulate the transcription rate of the gene and therefore the expression of the protease. Also, there is a tag SNP of the polymorphism rs243865 (-1306 C/T) and rs243866 (-1575G/A) [12], that are found in the gene promoter.

On the other hand, the genetic factor could interact with immunological and environmental factors to influence the individual susceptibility. A similar phenomenon could have been happening with the genotypical frequencies of the group of subjects with autoantibodies in comparison with HP seronegative subjects. The genotype GA+AA of rs11646643 in the *MMP2* gene shows a difference of 27.77% between these two populations, respectively. Curiously this genotype has been correlated to a difference between basal FVC/12 months through of linear regression.

This finding supports the participation of the rs11646643 genotype (GA+AA) in *MMP2* and its association in the lung function decline, independently of the development, or not autoantibodies.

Regarding MMP-2, this is involved in the remodeling of the extracellular matrix in the airways and pulmonary interstice, degrading molecules like collagen type I, III, IV, V, VII y X, gelatin, fibronectin, laminin and aggrecan [13].

In the comparison between the patient groups (HPAbs+ vs. HP), a higher plasmatic level of MMP-2 was identified in those with the GG genotype, suggesting that plasmatic levels were modified in a genotype-dependent mode. In our study, we found that the plasmatic levels of MMP-2 decrease in the subjects with genotype (GA+AA) of rs11646643 in comparison with the subjects genotype (GG). MMP-2 has the capacity of cleaving the chemotactic protein of the monocytes 3 (MCP-3) that allows getting to join to the chemotactic receptors 1, 2 and 3 without activating them. This characteristic suggests that the MMP2 decrease contributes to the persistent inflammatory response, not allowing the modification MCP-3 [14].

The variation in the levels of MMPs is affected by the course and progression of the disease; interestingly, the association seems to be in function of the genotype and not of the clinical phenotype.

A limitation in our study is that our Mexican mestizo population for ancestry has several genetic contributions, mainly Amerindians and Caucasians in different proportions.

Our result contributes to scientific knowledge, identifying candidate polymorphisms/genes in the development and progression of the HP. Nowadays we still require more assays to evaluate these variants like potential markers of clinical phenotypes for the disease, considering that HP is an inflammatory disease that in some people can develop chronic fibrosing disease, increasing morbidity and mortality.

## 5. Conclusions

Our findings demonstrate a subgroup of patients with hypersensitivity pneumonitis who develop autoantibodies without meeting the classification criteria for connective tissue disease. The presence of these antibodies contributes to a significant inflammatory response, possibly detonated by chronic exposure to a given environmental antigen.

Two polymorphisms in the *MMP1* and *MMP2* genes are associated with the risk of hypersensitivity pneumonitis in the Mexican mestizo population.

There are differences in the plasma levels of the MMP-2 protein among HP patients genotype-depending.

Our study suggests that polymorphisms and genes associated with metalloproteases influence the development of HP and worsening of lung function.

The identification of autoantibodies in patients with HP can have an impact on the course of the disease and the subsequent therapeutic management of interstitial lung diseases, since our results show, there are demographic and functional differences that probably influence prognosis.

## 6. Patents

None to declare.

## Figures and Tables

**Figure 1 biomolecules-09-00574-f001:**
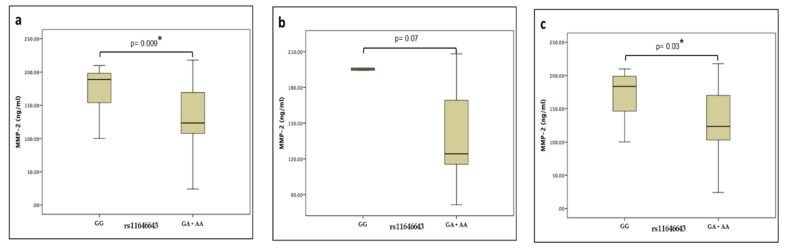
Median values and interquartile ranges of MMP-2 plasma levels in (**a**) genotypes of hypersensitivity pneumonitis (**b**) phenotype with positive autoantibodies (**c**) phenotype negative to autoantibodies.

**Table 1 biomolecules-09-00574-t001:** Characteristics of single-nucleotide polymorphisms (SNPs) included.

Gene	SNP	Chr Position	Allele Change	MAF *	Consequence/Gene Location
*MMP1*	rs470215	chr11:102790368	A>G	G = 0.31373	3’ UTR variant
	rs7125062	chr11: 102792772	T> C	C = 0.33799	Intron variant
	rs2071232	chr11:102794938	T>C	C = 0.1993	Intron variant
*MMP2*	rs243839	chr16:55495499	A>G	G = 0.2914	Intron variant
	rs243835	chr16:55502710	C>T	T = 0.4565	Intron variant
	rs243864	chr16:55478410	T>G	G = 0.1919	2 Kb Upstream variant
	rs11646643	chr16:55484965	A>G	G = 0.3101	Intron variant
*MMP9*	rs3918253	chr20:46010872	C>T	T = 0.42674	Intron variant
	rs3918278	chr20:46007015	G>A	A = 0.0218	2 Kb Upstream variant
*MMP12*	rs12808148	chr11:102862432	T>C	C = 0.1190	500 bp Downstream variant
	rs17368659	chr11:102872031	G>T	T = 0.1014	Intron variant
	rs2276109	chr11:102875061	T>C	C = 0.0988	2 Kb Upstream variant

* gnomAD: Allele frequencies are from The Genome Aggregation Database. Cite http://gnomad.broadinstitute.org/ Chr: Chromosome. MAF: Minor allele frequency. UTR: Untranslated region.

**Table 2 biomolecules-09-00574-t002:** Clinical and demographic characteristics.

Characteristics	HPAbs+ (*n* = 34)	HP (*n* = 104)	*p*-Value
Age, years	52.9 ± 9.3	50.9 ± 11.7	0.3
Sex, female. n (%)	29 (85.2)	89 (85.5)	1.0
BMI, kg/m^2^	27.3 ± 5.6	27.9 ± 5.2	0.5
Former smokers. n, (%)	5 (14.7)	30 (29.1)	0.01
Symptoms before diagnosis, months.	24 (1–120)	24 (6–192)	0.1
Antigen exposure			
Avian, n (%)	30 (88.2)	92 (88.4)	1.0
Unknown, n (%)	4 (11.7)	12 (11.5)	1.0
Diabetes mellitus, n (%)	6 (5.7)	10 (9.6)	0.2
Systemic hypertension, n (%)	3 (8.8)	27 (25.9)	0.002
Fever, n (%)	5 (14.7)	10 (9.6)	0.5
Digital clubbing, n (%)	7 (20.5)	26 (25)	0.4
Deceased, n (%)	6 (20)	17 (18)	0.9
≥10% FVC decline, n (%)	9 (26.4)	6 (5.7)	0.0001

HP: Hypersensitivity pneumonitis; Abs: Autoantibodies; mean ± SD; median (minimum and maximum values).

**Table 3 biomolecules-09-00574-t003:** Assessment of lung function and main laboratory findings in HP patients.

Characteristics	HPAbs+ (*n* = 34)	HP (*n* = 104)	*p*-Value
FVC % predicted	60 (29–97)	51 (20–98)	0.1
DLCO % predicted	60 (20–125)	41 (16–102)	0.06
pO2, mm Hg	50 (34.7–77.7)	47 (22–71.1)	0.1
Oxygen therapy, n (%)	9 (26)	37 (35)	0.09
PSAP, mm Hg	32 (20–77)	40 (20–90)	0.02
Laboratory blood test			
Optical density for avian antigen	1.45 (0.22–4.40)	0.89 (0.15–3.37)	0.03
White blood cell count, n x 10^3^ /mm^3^	8.1 (4.7–13.3)	8.1 (2.8–17.8)	0.6
Lymphocytes %	22.6 (12.5–36.5)	20.2 (3.8–77.9)	0.09
Eosinophils %	3.6 (1–12.3)	2.6 (1–18.8)	0.1
Hemoglobin g/dl	15.8 (13.2–20.9)	16 (11.7–21.2)	0.5
Hematocrit %	48.3 (39.4–63.3)	48.8 (35.8-68.7)	0.9
C-reactive protein mg/dl	1.023 (0.121–7.160)	0.541 (0.013–8.920)	0.006
BAL Lymphocytes %	54.5 ± 14.2	46.2 ± 21.2	0.03

HP: Hypersensitivity pneumonitis; Abs: autoantibodies; BAL: Bronchoalveolar Lavage. Mean ± SD; median (minimum and maximum values).

**Table 4 biomolecules-09-00574-t004:** Autoimmune serologic tests.

Characteristics	n (%)
ANA ≥ 1:320	17 (50.0)
Nuclear fine speckled	2 (5.8)
Nuclear coarse speckled	1 (2.9)
Homogeneous nuclear	7 (20.7)
Nucleolar	4 (11.7)
Fibrillar Cytoplasmatic	3 (8.9)
Others	17 (50.0)
RF ≥ 3x upper limit normal	5 (14.9)
Anti-topoisomerase (Scl-70)	2 (5.8)
Anti-Ro (SS-A)	1 (2.9)
Anti-La (SS-B)	2 (5.8)
Anti-dsDNA	4 (11.7)
Anti-CCP ≥ 3x upper limit normal	3 (8.9)

ANA: Anti-nuclear antibody; anti-CCP: Anti-Citrullinated Peptide Antibodies.

**Table 5 biomolecules-09-00574-t005:** SNPs and associated genotypes in the genes *MMP1* and *MMP2* in patients with hypersensitivity pneumonitis *versus* hypersensitivity pneumonitis with positive autoantibodies.

Gene/Model	SNP/ Genotype	Genotype Frequency (%)			HPAbs+ vs. HP		HPAbs+ vs. HC		HP vs. HC	
		HPAbs+ (*n* = 34)	HP (*n* = 104)	HC (*n* = 184)	*p*	OR (CI 95%)	*p*	OR (CI 95%)	*p*	OR (CI 95%)
*MMP1*	rs7125062									
Codominant	CC	20.59	25.96	73.37		1		1		1
	CT	47.06	49.04	2.17	0.3	1.21 (0.44–3.30)	0.0002	77.1 (20.3–292.6)	< 0.001	63.75 (21.25–191.20)
	TT	32.35	25.00	24.46		1.63 (0.54–4.85)		4.7 (1.7–12.8)		2.8 (1.53–5.45)
Dominant	CC	20.59	25.96	73.37						
	CT+TT	79.41	74.04	26.63	0.64	1.35 (0.52–3.46)	< 0.001	10.62 (4.34–25.96)	< 0.001	7.85 (4.54–13.57)
*MMP2*	rs11646643									
Codominant	GG	5.88	33.65	41.85		1		1		1
	GA	55.88	30.77	25.54	0.04	10.39 (2.2–48.1)	0.007	15.56 (3.46–69.85)	0.2	1.49 (0.82–2.73)
	AA	38.24	35.58	32.61		6.14 (1.2–29.2)		8.34 (1.81–38.38)		1.35 (0.76–2.40)
Dominant	GG	5.88	33.65	41.85						
	GA+AA	94.12	66.35	58.15	0.001	8.11 (1.83–35.84)	< 0.001	11.51 (2.67–49.49)	0.2	1.41 (0.85–2.34)

HPAbs+: HP patients with autoantibodies positive; HP: hypersensitivity pneumonitis patients without autoantibodies; HC: healthy controls.

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
