# Peer review of "MMP2 Polymorphism Affects Plasma Matrix Metalloproteinase (MMP)-2 Levels, and Correlates with the Decline in Lung Function in Hypersensitivity Pneumonitis Positive to Autoantibodies Patients"

_biomolecules, 2019, doi:10.3390/biom9100574_

Round 1

Reviewer 1 Report

The manuscript by Santiago-Ruiz et al. entitled “MMP2 polymorphism affects plasma matrix metalloproteinase (MMP)-2 levels correlates with the decline in lung function in hypersensitivity pneumonitis positive to autoantibodies patients” is a novel exploration of underlying SNP biology in a unique population suffering from hypersensitivity pneumonitis (HP).

The authors clearly explain the study design and their findings throughout the manuscript.  The data analysis and statistical methods employed are reasonable.  Even though the study is generally descriptive, the authors describe novel observations for the development of future studies.  The manuscript provides interesting associations underlying possible associations with SNPs in MMPs and the development of more severe HP.

Minor comments:

1.       Institutional review board (IRB), or similar regulatory/ethical board approval, should be described in the methods for appropriate tracking of the human based population study.

2.       The authors are encouraged to add SNP data for MMP9 and MMP12 in a format similar to that of Table 5.  Addition of this data in supplemental information would be sufficient.

a.       The other data concerning MMP9 and MMP12 in the current SI is appreciated, but a similar analysis of all MMPs should be presented.

3.       Associations based on the HapMap-Mex data should be more clearly detailed in the methods and results as these comparisons presumably underly the authors conclusion that “MMP1, MMP2, MMP9, and MMP12 are associated with genetic susceptibility for HP in Mexican Mestizo population.”

a.       The supplemental data tables contain the HapMap-Mex data, but no description in the main text directly references these data or analyses.

Author Response

The authors clearly explain the study design and their findings throughout the manuscript. The data analysis and statistical methods employed are reasonable.  Even though the study is generally descriptive, the authors describe novel observations for the development of future studies.  The manuscript provides interesting associations underlying possible associations with SNPs in MMPs and the development of more severe HP.

Thank you for your kind comments, the manuscript has been reviewed and corrected according to your suggestions.

Minor comments:

Institutional review board (IRB), or similar regulatory/ethical board approval, should be described in the methods for appropriate tracking of the human based population study.

That’s right, it was missing, the current version includes an appropriate description for ethical statements, including ethical board approval.

The authors are encouraged to add SNP data for MMP9 and MMP12 in a format similar to that of Table 5. Addition of this data in supplemental information would be sufficient. The other data concerning MMP9 and MMP12 in the current SI is appreciated, but a similar analysis of all MMPs should be presented.

Done, now supplementary information includes data for all genes/SNPs.

Associations based on the HapMap-Mex data should be more clearly detailed in the methods and results as these comparisons presumably underly the authors conclusion that “MMP1, MMP2, MMP9, and MMP12 are associated with genetic susceptibility for HP in Mexican Mestizo population.”

Thank you for your comments, now in the methods and results sections, populational data (HapMap-Mex) are described.

a. The supplemental data tables contain the HapMap-Mex data, but no description in the main text directly references these data or analyses.

Now in the results section, a paragraph describing differences in the frequencies is included.

Reviewer 2 Report

The manuscript entitled “MMP2 polymorphism affects plasma Matrix Metalloproteinase (MMP) -2 levels and correlates with the decline in lung function in hypersensitivity pneumonitis positive to autoantibodies patients” by Luis Santiago-Ruiz, et al. is a study of polymorphisms in the MMP1, MMP2, MMP9 and MMP12 genes in patients with hypersensitivity pneumonitis. The authors compared lung function and MMP-2 plasma levels between hypersensitivity pneumonitis patients with or without autoantibodies and healthy controls, and found that polymorphisms and genes associated with metalloproteases influenced the development of hypersensitivity pneumonitis and deterioration of lung function. This manuscript is well-organized and well-written. This article will be of interest to readers of biomolecules. I have a few comments on this manuscript.

Comments

The authors compare the results among 3 categorized group; hypersensitivity pneumonitis patients with or without autoantibodies and healthy controls. The readers would have interest in a direct comparison between “all” hypersensitivity pneumonitis patients (regardless of autoantibodies) and healthy controls. The authors should add description on the above analysis. The IRB approval is necessary and should be mentioned in the manuscript.

Author Response

The manuscript entitled “MMP2 polymorphism affects plasma Matrix Metalloproteinase (MMP) -2 levels and correlates with the decline in lung function in hypersensitivity pneumonitis positive to autoantibodies patients” by Luis Santiago-Ruiz, et al. is a study of polymorphisms in the MMP1, MMP2, MMP9 and MMP12 genes in patients with hypersensitivity pneumonitis. The authors compared lung function and MMP-2 plasma levels between hypersensitivity pneumonitis patients with or without autoantibodies and healthy controls, and found that polymorphisms and genes associated with metalloproteases influenced the development of hypersensitivity pneumonitis and deterioration of lung function. This manuscript is well-organized and well-written. This article will be of interest to readers of biomolecules. I have a few comments on this manuscript.

Thank you for your kind comments, the manuscript has been reviewed and corrected according to your suggestions.

Comments

The authors compare the results among 3 categorized group; hypersensitivity pneumonitis patients with or without autoantibodies and healthy controls. The readers would have interest in a direct comparison between “all” hypersensitivity pneumonitis patients (regardless of autoantibodies) and healthy controls. The authors should add description on the above analysis. The IRB approval is necessary and should be mentioned in the manuscript.

Thank you for your suggestion, now you can find detailed paragraphs in the results and discussion sections where comparations for “all” HP patients were done.

The IRB approval was missing, the current version includes an appropriate description for ethical statements, including ethical board approval.